# The Effects of Financial Stress and Household Socio-Economic Deprivation on the Malnutrition Statuses of Children under Five during the COVID-19 Lockdown in a Marginalized Region of South Punjab, Pakistan

**DOI:** 10.3390/children10010012

**Published:** 2022-12-21

**Authors:** Muhammad Babar Alam, Muhammad Shahid, Bashar Isam Alzghoul, Juan Yang, Rubeena Zakar, Najma Iqbal Malik, Asma Bibi, Kun Tang

**Affiliations:** 1Department of Public Health, Institute of Social and Cultural Studies, University of Punjab, Lahore 54000, Pakistan; 2World Health Organization, Peshawar 25000, Pakistan; 3School of Insurance and Economics, University of International Business and Economics (UIBE), Beijing 100029, China; 4Vanke School of Public Health, Tsinghua University, Beijing 100084, China; 5Respiratory Care Department, College of Applied Medical Sciences in Jubail, Imam Abdulrahman Bin Faisal University-Dammam, Jubail 35816, Saudi Arabia; 6Chinese Academy of Sciences and Technology for Development, Beijing 100038, China; 7Department of Psychology, University of Sargodha, Sargodha 40100, Pakistan; 8Independent Researcher, Lahore 54000, Pakistan

**Keywords:** COVID-19, financial stress, household deprivation, malnutrition, Pakistan

## Abstract

The lockdown after the COVID-19 pandemic not only caused public health crises and income stress but also put millions at risk of food insecurity and malnutrition across the globe, especially in low and middle-income countries [LMICs]. This study evaluated the effects of financial stress and household socio-economic deprivation on the nutritional status of 1551 children under the age of five during COVID-19 in Pakistan. A self-administered questionnaire was used between November 2020 and April 2021 to collect information on age, height, children’s weight, and socio-economic status from 1152 rural households from underdeveloped regions in Punjab, Pakistan. With the help of the proportionate simple random sampling method, this study employed a model (binary logistic regression) to calculate the likelihood of malnourishment. The findings showed that the stunting, underweight, and wasting prevalence rates during COVID-19 were 58.86%, 41.89%, and 8.11%, respectively, in the Bahawalpur region. According to the binary logistic regression analysis, among the household deprivation status (HDS) categories, the risks of childhood malnutrition were lesser in HDS-2 (OR = 0.05, 95% CI: 0. 005–0.879) and HDS-3 (OR = 0.04, 95% CI: 0.008–0.193). Similar to this, within the financial stress index (FSI) categories, the children in homes with medium financial stress had reduced odds of malnutrition (OR = 0.10, 95% CI: 0.018–0.567), and the children in households with low financial stress had reduced risks of malnutrition (OR = 0.006, 95% CI: 0.005–0.061). The proposed research found that stunting and underweight increased by 17.26% and 12.29% during the COVID-19 lockdown in the Bahawalpur region. Additionally, financial stress and socio-economic deprivation strongly affected children’s nutritional statuses during the COVID-19 lockdown in the Bahawalpur region of Southern Punjab.

## 1. Introduction

Pakistan has experienced unprecedented levels of financial stress and unemployment due to the COVID-19 pandemic. The poor families in deprived regions of Pakistan were severely affected by COVID-19 [1]. In poor families, the growth and nutrition of children under five were impacted by their pregnant or childbearing women due to severe financial stress during the lockdown. Financial stress significantly affects not only birth outcomes but also has a long-term impact on children’s nutrition status and overall family well-being. The COVID-19 lockdown affected people’s social and economic lives, restricted movement, and slowed global economic and financial activities, which had an immediate socio-economic effect on families [2,3]. Correspondingly, the health crisis evolved into a severe economic and social catastrophe with an impact on family income, unemployment, and well-being levels [4,5,6].

A lot of states were impacted by the COVID-19 pandemic, particularly those in southern Asia and Sub-Saharan Africa, where child undernourishment is a persistent problem [7]. Global experts believe that problems related to food and the economy during the time of the lockdown raised the probability of all forms of malnutrition [8]. During COVID-19, economic decline, food instability, and the suspension of regular nutrition programs may have increased the prevalence of wasting by 10–50% while also raising child mortality [9,10]. Undernutrition was high in the areas where COVID-19 was fatal [11]. It was estimated that in Canada, COVID-19 caused one in four children to experience food insecurity [12]. In the US, 93.5% of respondents reported having food insecurity in April 2020, and 41.4% of respondents claimed that COVID-19 caused them to consume fewer fruits and vegetables [13]. By 2022, the disruptions brought on by COVID-19 may have resulted in 9.3 million more wasted children, 2.6 million more stunted children, 168,000 child fatalities, and 2.1 million cases of maternal anemia and low-weight babies [14]. Additionally, USD 29.7 billion in productivity losses are expected due to increased stunting and child mortality. By stepping up dietary initiatives, an additional USD 1.2B per year would be required to minimize these consequences [14].

Evidence showed that the pandemic caused millions of families in LMICs to fall into poverty, which immediately worsened malnutrition. The number of children living in financially deprived homes has increased from 585 million before the outbreak to 702 million today [15]. Surveys from various countries have already revealed a sharp rise in the number of households with children experiencing food shortages that could be directly attributed to the pandemic [16,17]. Since the pandemic, multidimensional poverty has increased by 150 million more children [15]. In LMICs, COVID-19 not only causes infection but also compromises healthcare, food systems, humanitarian operations, and nutritional and educational initiatives [18,19]. According to UNICEF reports, mother and child nutritional services experienced a substantial reduction (75–100%) during lockdown periods and a 30% overall decline when the pandemic was not in lockdown [18,19]. Approximately 600 million children and families did not receive any government financial assistance as a result of COVID-19, according to a “Save the Children” report [15]. Obesity rates may increase as a result of a deteriorating diet and the social and physical constraints put in place by the government to stop the virus’s spread. The USA has expressed alarm about a potential rise in obesity rates [20]. Nearly 70% of disruptions in routine immunization [21] and 30% of the decrease in essential nutritional services (vitamin supplementation and nutrition promotion programs) were reported in LMICs [22]. Furthermore, efforts to treat children with severe wasting in 2020 were hampered [22]. In Pakistan, a 10% (PKR 1.1 trillion) economic loss in the 2021 financial year was estimated, along with a 33.7% increase in poverty [23]. According to the Pakistan Bauru of Statistics, the GDP growth rate was 6.03%, the unemployment rate was 4.35%, and inflation was 9.5% for the year 2021 (according to world bank data).

Financial stress is a psychological reaction to the feeling of instability, ambiguity, and danger in the management of financial assets and the decision-making process [24]. Financial stress impacted pregnant women more during COVID-19 in terms of the cost associated with prenatal and newborn care. In previous studies, financial stress was associated with mental health and depressive symptoms among women, as financial stress significantly affects the long-term offspring’s health and birth outcomes [25,26]. To date, it is not yet clear in Pakistan whether COVID-19 played any role in increasing malnutrition or not. Some studies conducted in other countries during COVID-19 concluded that COVID-19 hampered important public maternal-child health programs during lockdowns. So, it is hypothesized that the COVID-19 lockdown increased financial stress in families, and further financial stress and socio-economic deprivation have increased the malnutrition statuses in children under five. Therefore, considering this vast research gap, this study investigates the effect of financial stress and household deprivation status on the malnutrition statuses of children under five during COVID-19 in one of the most deprived regions of Punjab in Pakistan.

## 2. Theoretical and Conceptual Framework

The majority of studies on child malnutrition followed the framework of Becker (1965) and Strauss and Thomas (1995) [27,28]. The experts chose to pursue the household production function in doing so, assuming that every child resides in a structure known as a home and that the utility of each household maximizes as follows:*U = (L, X, N^i^)*(1)

In the given equation, home utility consists of the consumption of various items [*X*], leisure [*L*], and ensuring the nutritional quality [*N^i^*] of children. *N^i^* stands for anthropometric measurements such as stunting (HAZ), underweight (WAZ), and wasting (WHZ). Furthermore, the nutritional statuses of children [*N^i^*] depend on many socio-economic factors such as:*N^i^ = n [H, Z, W, C, ε]*(2)

Here, *N^i^* is a child’s typical anthropometric measurement, *C* stands for consumption, the *W*-vector stands for certain children, and the *H*-vector stands for the specific social and economic characteristics of a home. The health factor vector is represented by *Z*, while the children-specific error term is represented by the equation. Equation 2 captures the overall effect of child health in terms of the determinants of child health.

The reduced defined production function for the proposed malnutrition study is:*CIAFi = f(household socio-economic factor, factor specific to child-mother, individual and disease factors, εCIAF)*

The conceptual framework by Victora et al. (1997) was followed in this investigation. This concept divides variables into three categories: proximal or individual considerations, socio-economic reasons, and intermediate factors containing maternal and environmental concerns. Briefly, these factors may have an impact on the nutritional health of preschool children [29]. The study’s visual representation is presented in Figure 1**.**

## 3. Methods

### 3.1. Participants in the Study and the Eligibility Criteria

The anthropometric data (the height in relation to age measurement, the weight in relation to height measurement, and the weight in relation to age measurement) of 1551 children under five were examined in this study. Based on biologically implausible Z-score levels (less than -5 standard deviation and greater than +5 standard deviation), the World Health Organization defined the anthropometric outliers.

### 3.2. Research Area, Sample Design, and Data Collection

The primary data for this study were acquired through a simple proportionate purposive simple random sampling method from the rural parts of the Bahawalpur division through a self-administered questionnaire. The sample was distributed proportionally among 3 districts, 14 sub-districts (tehsils), and 42 union councils. In the Bahawalpur division, there were three districts (Rahymyar Khan, Bahawalpur, and Bahawalnagar), and in these districts, there were 14 tehsils. We selected all rural households in three districts as the sample frame, and among these three districts, the rural households in all 14 sub-districts or tehsils were used as the sample frame in our study. The names of the districts and tehsils are provided in Table 1. Furthermore, a total of 129 UCs were included in all four tehsils of the Rahimyar Khan district. There was a total of 88 UCs in all five tehsils of the Bahawalpur district, and a total of 135 UCs were in all five tehsils of the Bahawalnagar district. According to the 2017 census, approximately 75.36% of the population in the Bahawalpur division lived in rural areas, and the number of rural UCs was more than the number of urban Ucs. Therefore this study was conducted in the rural areas of the Bahawalpur division. Due to the cost and time constraints, we possessively selected only three rural union councils from each tehsil, and in this way, a total of 42 union councils were considered for this study. The fixed 384 samples of each district were proportionately distributed from the district to the tehsils and from the tehsils to the UCs at the first stage. The households were selected randomly from the UCs following the lady health worker’s register record in the second stage. All three sample designs for the districts of the Bahawalpur division were based on the probability proportionate to size (PPS). All rural households in the division were included in the sampling frame. The rural households participating in the survey were picked at random from the records of the lady health workers (LHWs). If more than one family lived on the same property or if joint families were living in the same home, the survey treated them as nuclear families if they produced their own meals. Female healthcare workers who had obtained the necessary training before being tasked with conducting anthropometric measurements took the measurements.

The data were collected in the research area for five months, from November 2020 and April 2021. Two days before the data collection, the lady health professionals informed the women and their immediate relatives about the study’s purpose in their regional languages (Saraiki and Punjabi) to obtain their verbal agreement and expression of interest in taking part in the research. A total of 1152 mothers accepted to become part of the study willingly, and the mothers verbally consented at the time of the pre-interview meeting. Due to cultural restrictions and the fact that the majority of the mothers lacked formal education, written consent was not acquired. The sample size for each district was determined to be *n* = 384 households, with the confidence interval and confidence level set at 5%. This 384 sample size was kept fixed for each district in the Bahawalpur division. Thus, for the three districts, the total sample size came to 1152 respondents. The following information is provided, which determined the sample size:Sample size = Z ^2^ ∗ (p) ∗ (1 − p)/c ^2^ Here:

Z = value of Z (i.e., 1.96 for a 95% confidence level), p = decimalized percentage of a choice’s selection (for the required sample size, we used 0.5), c = confidence interval in decimal form (e.g., 0.04 = ±4)
Size of Sample = (1.96) ^2^ ∗(0.5) ∗ (1 − 0.5)/(0.05) ^2^ = 384.16 = 384
Total sample of the Bahawalpur division = (Rahim Yar Khan district 384 + Bahawalpur district 384 **+** (Bahawalnagar district 384) = 1152

Table 1 shows the proportional distribution of each district to the council unions (N = 384 in tehsils):NI = n ∗ Ni/N

The calculation for the formula for each UC sample = UCs 1–3 population/3 council unions total population*tehsil sample size, NI = in each union council, the number of sampled respondents, I = number of UCs in the research area, i.e., 1, 2, 3…, 42; n = sample’s total size.

### 3.3. Outcome Variable

In the study, the DV (CIAF) was dichotomized into two groups: “1”: an undernourished child, and “0”: not an undernourished child. A implies no failure, B means only stunted, C means only wasting, D means only underweight, E means both stunted and underweight, F means both wasting and underweight, and G means underweight, stunting, and wasting. These seven groups are used by the CIAF to categorize children. The CIAF assesses the prevalence of childhood malnutrition. Except for group A, all groups are added together to provide the overall estimate of child malnutrition prevalence. Following WHO’s standard guidelines for child growth (2009), the CIAF (outcome variable) was grounded on three indices, WAZ, WHZ, and HAZ, and was then followed by anthropometric measurements [30].

### 3.4. Measures

The Development of Household Deprivation Status: The household deprivation status index was employed in the proposed study used in India [31]. Recently, this index was also used in South Punjab, Pakistan [32]. The HDS index employs six variables: 1: the type of household (cemented or mud), 2: whether the household owns any land, 3: the presence or absence of an electricity facility in the house, 4: the presence or absence of a drinking fountain inside the home or household, 5: whether or not any one member of the family is literate, and 6: whether a newspaper, TV, or radio is kept in the house. These are binary in nature. The sum of all six variables yields overall scores, which vary from zero to six. Those without any of the six variables or only one or two items are encompassed in HDS-1 and are referred to as “MD”, which means moderate deprivation, indicating disadvantaged parts of the community. “JAD” means just above the deprivation and refers to persons who have three HDS-2s. Those with four or six items on the HDS-3 indicate “WAD”, which means well above the deprivation. The HDS does not directly evaluate a household’s financial circumstances, such as the total spending, per capita income, or the index of living standards; nevertheless, it broadens the measure to show households above the three dimensions as disadvantaged [31].

The Development of the Financial Stress Index During COVID-19: We constructed the FS index using existing research from PubMed and Google Scholar and also focused on the current COVID-19 situation, local study area, and study theme. We only reviewed papers examining the association between financial stress and malnutrition/health/COVID-19. The primary focus remained on research that used financial stress indicators or indexes and was further linked to health/malnutrition/COVID-19 issues.

We discovered 18 publications that directly addressed the association between financial stress and malnutrition/health/COVID-19. For a literature search, the following keywords were used: financial stress and malnutrition, (2) financial stress and COVID-19, (3) financial stress and health, and (4) the financial stress index and health.

Most of the studies concluded that financial stress arose due to the COVID-19 pandemic, and due to financial stress, people suffered in terms of fulfilling their nutritional needs, and the impact of household financial deprivation and stress significantly shifted from the family to the children. In the literature, the authors discovered that the information in the majority of studies on the financial stress indexes and variables were related to job losses in COVID-19, reductions in salaries, business losses or reduced business activities, sold assets to afford basic amenities of life, and borrowed money to meet expenses during COVID-19. We discussed this with a few experts on the subject after searching the relevant literature. All of the experts held doctoral degrees in psychology and public health. According to them, the financial stress tool should be simple, short, and easy to grasp for local respondents, with more focus on the current COVID-19 situation. As a result, the study integrated basic questions to create the financial stress index, which covered the essential variables that a typical respondent faced throughout COVID-19.

One of the important variables in this study was the financial stress index, which was produced as part of the study. Ten questions on respondents’ financial stress during COVID-19 that comprised this index were as follows: was there a: Reduced salary from the employer? Complete job loss? Business loss? Reduced business or work activities? Delay in receiving income? Reduction in saving or a dependency on saving? Reduction in remittances sent by family members from abroad or within the country? Situation of sold assets to afford basic amenities of life? Situation of sold livestock to afford the necessities of life? Situation of borrowing money from others to afford your family expenses?

The index is binary in nature; if the respondent answers “yes” to the question, it assigns a value equal to one; otherwise, it assigns a value equal to zero. This signal is further classified into three levels of stress: low, medium, and high. For the low financial stress group, the value range is 0–3, for the medium financial stress group, it is 4–7, and for the high financial stress group, it is 8–10. A high value indicates that the respondent, who represents a household, is under significant financial strain as a result of COVID-19.

### 3.5. Statistical Analysis

The study observed the impact of financial stress and household deprivation on the malnutritional status (CIAF) of children under five in the Bahawalpur region. The data were cleansed of any outliers prior to examining the connection between malnutrition and the explanatory factors. The Z-scores from the data that fell outside of the WHO flags were also disregarded. A total of 930 children were included in the study for analysis, while 621 children out of a total of 1551 children under the age of five were skipped from the analysis because the data were outside of the accepted range (less than −5 and more than +5). For each explanatory variable, descriptive statistics were used to determine the prevalence of CIAF (malnutrition). The logistic regression technique was used to evaluate the impact of various socio-economic characteristics, especially household deprivation status and financial stress, on the malnutritional statuses of children. The logistic regression model employed three levels of significance to demonstrate the significant link between the IV and the outcome variable, i.e., p 1%, p 5%, and p 10%. The STATA 15 statistical software was used for the data analysis.

The method of logistic regression divides the likelihood of malnourishment into two categories: “1” for malnourished children and “0” for healthy children. The premise was that a wide range of socio-economic determinants influences the status of childhood malnutrition. According to the socio-economic indicators, the logistic regression model calculates the likelihood of the outcome variable (child malnutrition). The DV or outcome variable was configured in binary nature (0 or 1). The specification form for the binary logistic regression is shown underneath:*P (Y_i_ = 1|X_1i_, X_2i_…, X_kn_) = F (β**_0_**+ β**_1_**× _1i_ + β**_2_**X_2i_ + … + β_n_X_kn_)*

In this equation, the dependent variable (CIAF) is denoted with the symbol *Y_i_*, which stands for the indicators of child malnutrition. *X* signifies the explanatory variables, *β*’s means the coefficients of interest, which describe how strongly the dependent variables are associated. Here, the dependent variable is *Y*, and if *Y_i_ = 1*, this means a malnourished child, whereas if *Y_i_ = 0*, the child is not malnourished. *X= (X_1i_, X_2i_ …, X_kn_)* are the IVs, and the independent variable’s observed value for observation *i* is *xi*.

### 3.6. Ethical Consideration

The departmental doctoral program coordination committee (DDPCC) of the University of Punjab Lahore authorized the research on 25 June 2020. The study’s instruments and methods were also examined and approved by the Institute of Social and Cultural Studies (ISCS). Before obtaining the data, the health officials, mothers, and LHWs were informed of all the study’s specifics, and oral and informed consent was obtained from all of them.

## 4. Results

### 4.1. Distribution of Data

A total of 1551 children had their anthropometric measures collected. Following the anthropometric evaluation, it was necessary to examine the data distribution. In the distribution, Z-scores between −1 SD and >−2 SD indicated the normal category, Z-scores between −2 SD and >−3 SD showed moderate malnourishment, and Z-scores between −3 SD and >−5 SD showed severe malnourishment. The WHO does not take Z-scores over 6 SD into consideration. Figure 2A–C shows that while there is very little evidence of wasting (WHZ), there are deficits in HAZ (stunting) and WAZ (underweight).

Figure 3 displays the graphical relationships between the various anthropometric indicators. Figure 3A–C shows that while no association is present between HAZ and WAZ, there is a weakly positive relationship between WAZ and WHZ, as well as HAZ and WHZ.

### 4.2. Descriptive Statistics

We calculated the prevalence rates for wasting, underweight, and stunting, which were 8.11%, 41.89%, and 58.86%, respectively, in the Bahawalpur region. In comparison, the overall malnutrition (CIAF) prevalence was 63.23%. Table 2 displays the malnutrition prevalence rates according to the child’s gender. In male children, the wasting and underweight rates were higher, while the stunting rates were higher in female children. Table 3 illuminates the frequency of malnourishment in children according to age (in months). The table depicts that the HAZ, WHZ, and WAZ rates are high in children aged 13–48 months.

Table 4 shows that during COVID-19, the stunting and underweight rates significantly increased from the provided rates before COVID-19 in the Bahawalpur division. Moreover, the Bahawalpur region had a high malnutrition prevalence in the region of Punjab. Table 5 shows the percentage of children that experienced CIAF (malnutrition) in relation to several explanatory variables.

Figure 4 shows the connection between household deprivation status (HDS) and child nutritional status. The HDS (1 and 2) is the most impoverished in society. The stunting, underweight, and wasting prevalence rates fall as the household moves from HDS 1 to 2 to 3. It shows that the increase in the basic necessities of life in households reduces the malnutrition prevalence in children.

Figure 5 shows the association between financial stress (FSI) and child malnutritional status. The figure depicts that the prevalence rates of wasting (WHZ), stunting (HAZ), and underweight (WAZ), decreased when the household shifted from high-stress to medium-stress and then low-stress continuously.

### 4.3. Estimates of Logistic Regression

Table 6 displays the CIAF logistic regression estimates. According to the findings, male children had a lower risk of malnutrition than their female counterparts (OR = 0.72, 95% CI: 0.52–0.98). Low birth weight was linked to an increased danger of malnourishment in children between the ages of 13 and 36 months. Lower likelihoods of malnourishment were associated with birth order/interval of children aged four to five years (OR = 0.47, 95% CI: 0.31–0.71). The risks of malnutrition in children were higher in mothers with BMIs higher than 18.5 kg/m2 (OR = 1.45, 95% CI: 0.95–2.21). Malnutrition among children under the age of five was more likely to occur in mothers without any employment or jobs (OR = 3.14, 95% CI: 0.81–12.18). Fathers with private employment or their firm had a lower likelihood of malnutrition in their children under five.

The risks of children being malnourished were lower across all household deprivation status (HDS) categories in the HDS-2 (OR = 0.05, 95% CI: 0. 005–0.879) and HDS-3 (OR = 0.04, 95% CI: 0.008–0.193) categories. According to the financial stress index (FSI) categories, the children belonging to households with medium financial stress had less danger of malnutrition (OR = 0.10, 95% CI: 0.018–0.567), and the children from households with low financial stress had less danger of malnutrition (OR = 0.006, 95% CI: 0.005–0.061).

## 5. Discussion

This study constructed an index of financial stress created by the COVID-19 lockdown in Pakistan. Furthermore, it observed the impact of financial stress and household socio-economic deprivation during the COVID-19 lockdown on the nutritional statuses of children under five in one of the marginalized regions of South Punjab, Pakistan. The results of the study depicted that the prevalence of underweight children was 41.89%, and the prevalence of stunted children was 58.86% in the Bahawalpur division during the COVID-19 lockdown. Furthermore, the descriptive results illustrated that 2.55% and 88.7% of malnourished children belonged to the HDS-1 and HDS-2 households category, which shows the deprived segment of society. Moreover, 91.84% of malnourished children belonged to households that faced high financial stress during the COVID-19 lockdown in the Bahawalpur region.

The logistic results showed that male children had a lowered likelihood of becoming undernourished than their female counterparts. The results also depicted that the age of the children was linked with a higher malnutrition probability. Several studies argued that, in general, male children are preferred by their parents, and girls are more deprived than men, especially in terms of food and nutrition, so due to gender discrimination at home, girls are more likely to be malnourished than male children. [33,34]. However, some studies argued that male children are at higher levels of malnourishment compared to their female counterparts due to the poor socio-economic status of the household [35,36]. The preference for sons in poor societies is usually because sons are usually the source of income for the family, but females are a burden [37]. The injustice in food distribution is also present in disadvantaged houses in which mothers first feed their husbands and sons and then offer it to their daughters, and sometimes poor quality and deficient foods are left for the daughters [38]. Poverty in rural as well as deprived urban areas is higher in Pakistan, and girls and women are victimized in their nutritional intake and health outcomes [39].

Healthy mothers deliver healthy offspring. In the worst situation, a mother with a low BMI who is malnourished delivers a low-weight infant, and there is a high likelihood that those children may experience starvation during infancy [40]. Low BMIs in mothers (18.5 kg/m^2^) raise the risk of low-birth-weight infants [40]. The findings of our study demonstrate a substantial relationship between mothers’ BMI and the prevalence of malnutrition in children under the age of five. Our results do not agree with a previous investigation [41].

There may be a trade-off between a woman’s ability to work and her ability to provide for her children. If women work properly, child growth and care, particularly child nourishment, might be impacted because of the reduced time spent with the child owing to work [42]. However, women’s employment status also makes a separate contribution to children’s nutrition. The household’s ability to purchase food, maintain a healthy diet, and pay for the necessities of life is increased by the income of women. The results of the study explain that the mother's employment status was significantly associated with the child's malnutrition status. The findings of a study showed that working women in Pakistan’s households in the first two quintiles of the wealth index, the poorer and poorest, did not contribute to the nutritional status of the child. However, in the third and middle quintiles, working women did [43]. While child care should not be sacrificed, women’s employment should be sufficient to maintain the household’s socio-economic standing. During the COVID-19 pandemic and lockdowns, low work or employment conditions reduced incomes vital to nutrition and food security.

The findings show that the higher the birth order of children, the lower the likelihood of malnourishment. A study from Bangladesh also had similar results revealing that decreased chances of malnutrition were associated with children’s birth order when they were 4–5 years old [40]. However, it is likely that it does not rise linearly with birth order. Because natural foods, such as fruits, milk, and vegetables, are affordable and accessible in rural areas, it may be a reflection of the reality that the majority of parents were meeting their children’s basic nutritional needs. According to a study conducted in Nepal, severe acute malnutrition was significantly correlated with birth intervals of less than 24 months [44]. The same research showed that greater birth order dramatically enhanced the risk of stunting in Pakistan [45].

The results show that the fathers’ income status and household deprivation status were correlated with malnutrition status. In India, prior research has shown that household deprivation affects child nourishment in the destitute parts of society in comparison to households with access to more necessities [46]. According to a different study conducted in India, more than half of truly impoverished households have at least one child who is stunted or underweight when compared to their non-poor counterparts. This study used a relatively comparable assessment of household asset deprivation [47]. The likelihood of malnutrition was found to be higher in the most deprived index than in the least deprived in a similar study conducted in the UK. This study used a household deprivation index that includes six items: employment, income, health deficiencies, disabilities, skills and training, education, housing, and access to services. Additionally, those with medium-high malnourishment risks reside in neighborhoods with more deprived households [48].

Socio-economic inequality causes social injustices that ultimately result in hunger. According to research from India, one of the main causes of socio-economic inequality is the wealth status in children’s undernutrition [48]. In the Ecuadorian highlands, newborn malnutrition was highly correlated with socio-economic class [49]. According to a study including an HDS index based on housing, location, employment rates for men, car ownership, and access to low-quality social amenities, the malnourishment rates were greater in less privileged homes, where they were 6.9 % vs. 9.5 % in the most fortunate ones [50]. Additionally, in Kenya, a study found that children’s nutritional quality was significantly impacted by their household assets [51]. Research conducted in Pakistan also revealed that household social and economic disadvantage was a significant cause of child malnutrition. Recent literature in Pakistan highlighted that richer households have a lower probability of stunting and wasting prevalence [52]. A case study from the deprived region of Punjab Pakistan revealed that malnutrition chances decrease with an increase in wealth status from rich to the richest. The study revealed that as the household shifts from a lower socio-economic status (HDS-1) to a middle socio-economic status (HDS-2) and further to a rich socio-economic status (HDS-3), the rates of underweight and stunted children decrease [53].

The logistic results depicted that the malnutrition prevalence was higher among those children whose families have a high financial stress index. These studies concluded that parents are a crucial group of interest since parental financial distress could harm family life and children’s psychosocial development [54]. According to a study from Switzerland, even though they are less directly affected by the virus, children are paying a heavy price for the crisis’ indirect impacts, such as poor eating, effects on mental health, social isolation, addiction to screens, and a lack of access to education and healthcare, especially among marginalized communities [55].

A study depicted that perceived stress scale scores increased during the outbreak from low to moderate levels of stress, and some financial stresses, including job losses, were also linked to more severe depressive symptoms and COVID-19-related anxiety [56]. Another study illustrated that financial and psychological stressors related to COVID-19 were significantly linked with greater alcohol-related problems and different kinds of drinking behaviors [57]. Previous studies indicated that the economy and health are more at risk from the COVID-19 pandemic. Due to the outbreak, 11% of Canadian respondents lost their jobs, while an additional 8% saw income reductions [58]. Another Canadian study found that juggling employment with childcare and homeschooling and financial instability are the main factors affecting family stress during COVID-19. Studies from central and southeastern Europe and China indicated that financial stress and related constructs (for example, money worries, financial anxiety, and worry) demonstrated a major impact on an individual’s quality of life [59,60]. Not having enough money to participate in the same activities and anticipating having more student loan debt were two of the top financial concerns for college students, according to research from Ohio, USA [61]. Another study from the USA indicated that financial stress and anxiety among workers could occasionally emerge as physical problems such as backaches, headaches, high blood pressure, and ulcers. Money problems have in the past been indirectly linked to employee performance, attrition, productivity, and healthcare expenditures [62]. The COVID-19 pandemic resulted in financial hardship for 43% of participants in the USA, and this financial stress was significantly linked to an elevated risk of a clinically significant depression score, including income [63]. Respondents in another study reported that household finances were the main source of financial stress along with an immense amount of pressure to support their families during COVID-19 in Latino Americans [64]. A multivariate analysis from western Sydney, Australia, indicated that financial hardship factors, particularly job and housing security issues and the inability to pay mortgages or rents, established the highest connections with psychological suffering [65]. Studies from the UK have discovered a link between poorer mental health and having financial troubles, as well as worrying about money and having debt [66,67], as the COVID-19 pandemic contributed significantly to families’ financial stress [68]. Findings from Wales revealed that the COVID-19 pandemic aggravated mental health issues in children who were already at risk, and the negative effects of financial stress (e.g. the inability to pay expenses, loss of income, or job loss) were connected to the mental health of parents [69].

### Strengths and Limitations of the Study

The study assessed the impact of household socio-economic deprivation and financial stress on child malnutrition which was specifically shaped by COVID-19 during the lockdown in rural areas of Southern Punjab, Pakistan. The data were collected from the rural areas of three districts with 1152 households. So, for policy purposes, the results and scope of this study could be generalized to the whole of rural Punjab, Pakistan. The limitation of this study is that the questions developed to design the financial stress index were related to stress and job loss issues that arose during COVID-19. Moreover, this study used a prearranged set of variables because of the importance of time to meet the objectives. The future directions could be how COVID-19 disrupted essential maternal and child health and child nutrition programs during the lockdown. Moreover, researchers could also conduct a post-COVID-19 analysis for the malnutrition prevalence and determine the factors which played a role in overcoming malnutrition in post-COVID-19.

## 6. Conclusions

The study assessed the effect of financial stress and household socio-economic deprivation on the nutritional statuses of children under five during the COVID-19 lockdown. The results of the study depicted that the prevalence of underweight children was 41.89%, and the prevalence of stunting was 58.86% in the Bahawalpur division during the COVID-19 lockdown. The logistic regression results depicted that gender, birth order, and age of the child, BMI and working status of the mother, and father or parental working status were the other significant determinants of child malnutrition status. Moreover, the results of the policy variables of our study (financial stress and household deprivation status) showed that the probability of the child being malnourished decreased with the rise in the social and economic status of the household. Similarly, the probability of child malnutrition decreased when a household shifted from high financial stress to medium financial stress and then low financial stress. In sum, our findings concluded that household socio-economic deprivation and financial stress created by the COVID-19 lockdown impacted the nutritional status (an increase of 17.26% stunted children and 12.29% underweight children). The socio-economic deprivation in downgraded areas contributed to malnourishment, which might be eliminated with equal possibilities for human growth and increased funding for disadvantaged groups in less developed areas in the form of social safety nets. Moreover, providing job security, especially during unexpected situations, such as COVID-19, would not only help people to feed their families but also overcome mental health issues.

## Figures and Tables

**Figure 1 children-10-00012-f001:**
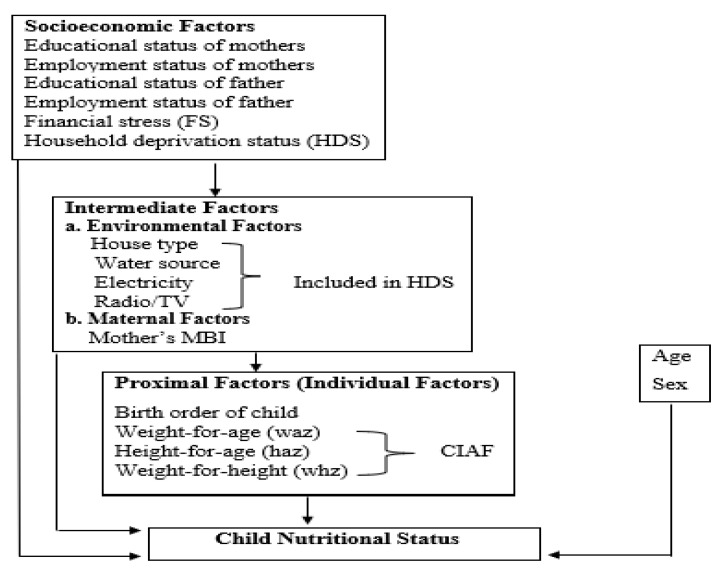
Study’s conceptual framework for the child malnutrition determinants.

**Figure 2 children-10-00012-f002:**
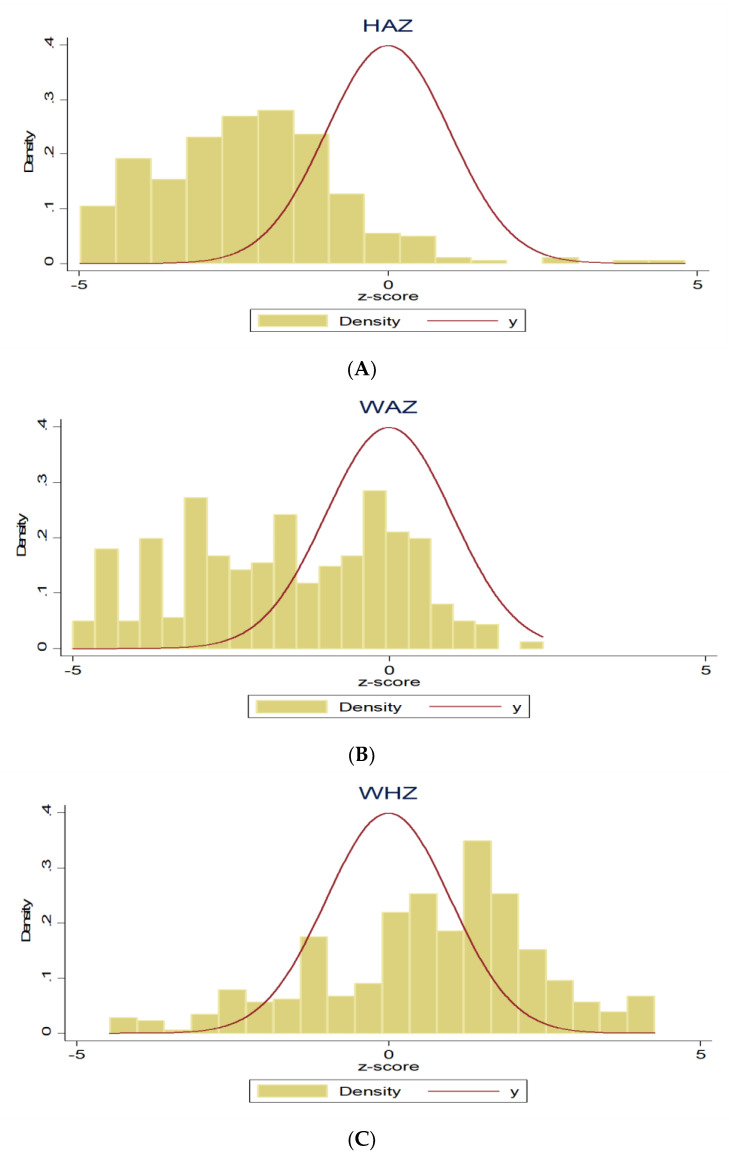
(**A**) Distribution of the Z-Scores for HAZ, (**B**) distribution of the Z-Scores for WAZ, and (**C**) distribution of the Z-Scores for WAZ.

**Figure 3 children-10-00012-f003:**
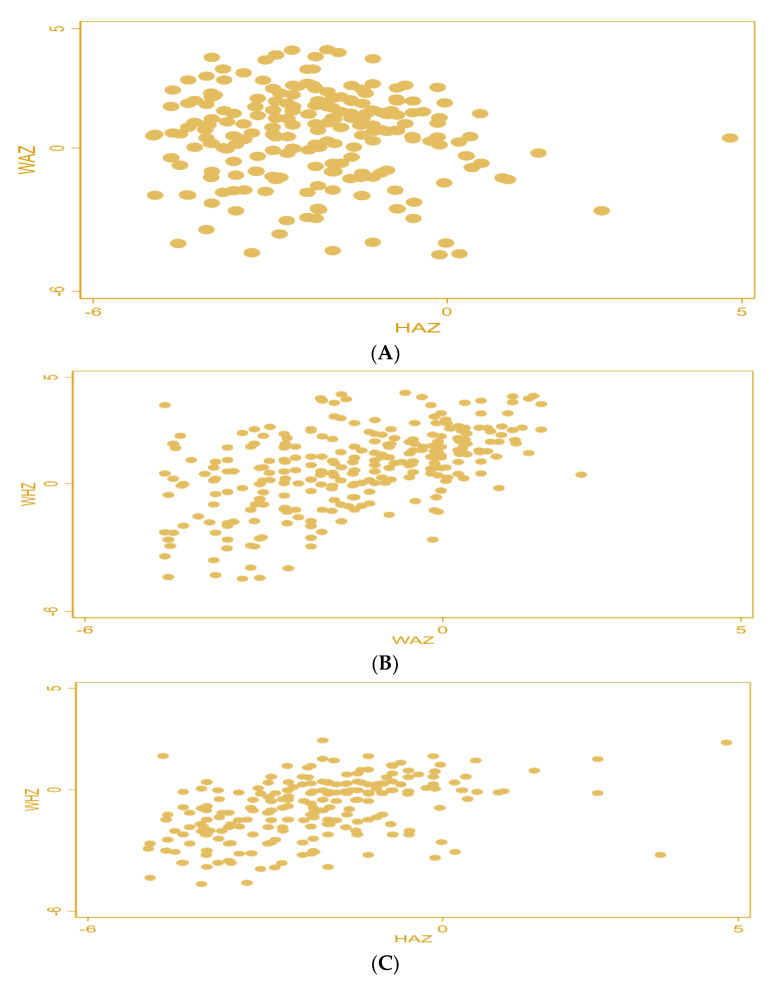
(**A**) Correlation between the stunting and underweight variables, (**B**) correlation between the wasting and underweight variables, and (**C**) correlation between the stunting and wasting variables.

**Figure 4 children-10-00012-f004:**
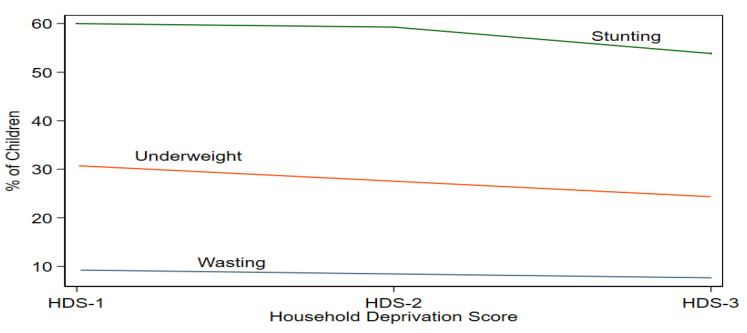
Malnutrition Prevalence by Household Deprivation Status. Source: author.

**Figure 5 children-10-00012-f005:**
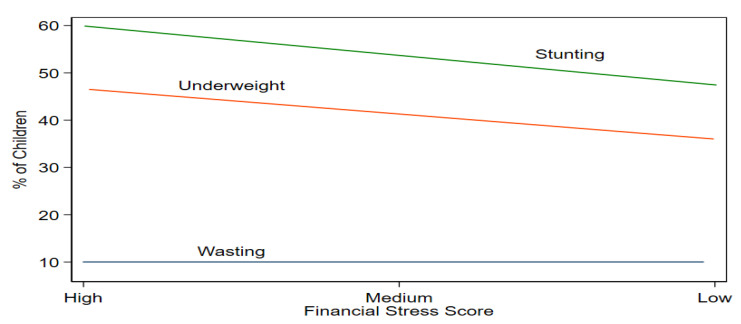
Malnutrition Prevalence by Financial Stress. Source: author.

**Table 1 children-10-00012-t001:** Sample size proportional distribution from the sub-districts to the council unions.

Division /Region	Districts	Tehsils	Council Union Number	Proportionate Sample/HH
Bahawalpur	1. Rahimyar Khan District Sample = 384 (100%)	Khanpur Sample = 96	UC1	26
			UC2	34
			UC3	36
		Liaquatpur Sample = 81	UC4	25
			UC5	26
			UC6	30
		Rahimyar Khan Sample = 115	UC7	34
			UC8	46
			UC9	35
		Sadiqabad Sample = 92	UC10	33
			UC11	32
			UC12	27
	2. Bahawalpur District Sample = 384 (100%)	Bahawalpur Sample = 111	UC13	29
			UC14	39
			UC15	43
		Hasilpur Sample = 65	UC16	27
			UC17	11
			UC18	27
		Khairpur Tamewali Sample = 54	UC19	18
			UC20	23
			UC21	13
		Yazman Sample = 77	UC22	10
			UC23	41
			UC24	26
		Ahmad Pur East Sample = 77	UC25	36
			UC26	20
			UC27	21
	3. Bahawalnagar District Sample = 384 (100%)	Bahawalnagar Sample = 100	UC28	37
			UC29	48
			UC30	15
		Chishtian Sample = 92	UC31	20
			UC32	54
			UC33	14
		Fort Abbas Sample = 54	UC34	7
			UC35	23
			UC36	26
		Haroonabad Sample = 73	UC37	55
			UC38	12
			UC39	7
		Minchinabad Sample = 65	UC40	10
			UC41	27
			UC42	29
	Districts = 3	Tehsils = 14	Total UCs = 42	HH = 1152

Source: the authors.

**Table 2 children-10-00012-t002:** Malnutrition prevalence according to the sex of the child.

Indicators	Stunting (n = 948)		Underweight (n = 1368)		Wasting (n = 1221)		CIAF (n = 930)	
	N	%	N	%	N	%	N	%
Male	264	47.31	300	52.36	57	57.58	279	47.45
Female	294	52.69	273	47.64	42	42.42	309	52.55
Total	558	58.86	573	41.89	99	8.11	588	63.23

Source: the authors.

**Table 3 children-10-00012-t003:** Prevalence Of Malnutrition in Children according to Age (In Months).

Indicators	Stunting (n = 948)		Underweight (n = 1368)		Wasting (n = 1221)		CIAF (n = 930)	
	N	%	N	%	N	%	N	%
0–12 months	51	9.14	105	18.32	18	18.18	57	9.69
13–24 months	78	13.98	144	25.13	24	24.24	78	13.27
25–36 months	171	30.65	165	28.80	18	18.18	180	30.61
37–48 months	141	25.27	105	18.32	27	27.27	153	26.02
49–60 months	117	20.97	54	9.42	12	12.12	120	20.41
Total	558	58.86	573	41.89	99	8.11	588	63.23

Source: the authors.

**Table 4 children-10-00012-t004:** Comparison of the Stunting, Wasting, and Underweight prevalence before and during COVID-19 in the Bahawalpur division.

	Stunting	Underweight	Wasting
Pakistan	38%	23%	8%
Punjab	31.5%	21.2%	7.5%
Bahawalpur Division Before COVID-19	**41.6%**	**29.6%**	**8.3%**
Bahawalpur Division During COVID-19	**58.86%**	**41.89%**	**8.11%**

Source: the national rates (PDHS-2017-18), provincial and Bahawalpur division rates before COVID-19 (MICS- 2017-18), and during COVID-19 rates are calculated through the study surveyed data (collected from July 2020 and November 2020).

**Table 5 children-10-00012-t005:** Descriptive analysis concerning the relationship between several socio-economic factors and CIAF/malnutrition (N = 930).

Variables	Categories	*F*	%	*p*-Values
Child Gender	Male	279	47.45	0.011 ***
	Female	309	52.55	
Child Age (Months)	0 to 12	57	9.69	0.000 ***
	13–24	78	13.27	
	25–36	180	30.61	
	37–48	153	26.02	
	49–60	120	20.41	
Number of Order of Birth	Birth order 1	165	26.53	0.000 ***
	2 or 3	237	40.31	
	4 or 5	123	20.92	
	6 or above	72	12.24	
Mother BMI	Less than 18.5 kg/m^2^	78	13.27	0.014 ***
	Equal to or more than 18.5 kg/m^2^	510	86.73	
Education of Mother	No education	450	76.53	0.000 ***
	Primary	66	11.22	
	Middle	42	7.14	
	Matric	18	3.06	
	FA and higher	12	2.04	
Education of Father	No education	405	68.88	0.489
	Primary	105	17.86	
	Middle	21	3.57	
	Matric	48	8.16	
	FA and higher	9	1.53	
Working Status of Mother	Working	9	1.53	0.000 ***
	Not working	579	98.47	
Working Status of Father	Govt. job	24	4.08	0.001 ***
	Private job	25	4.09	
	Own business	75	12.76	
	Labor or daily wages and agriculture	465	79.08	
Household Deprivation Status (HDS)	HDS-1	15	2.55	0.017 **
	HDS-2	552	88.78	
	HDS-3	51	8.67	
Financial Stress	High-stress	540	91.84	0.000 ***
	Medium-stress	33	5.61	
	Low-stress	15	2.55	

Source: the authors’ estimation [level of significance: ** if *p* < 0.05, *** if *p* < 0.01].

**Table 6 children-10-00012-t006:** Results for child malnutrition and its correlations from the binary logistic regression analysis.

Variables	Categories	Odd. Ratios	*p*-Vales	CI: 95%
Child Gender (Female-reference)	Male	0.72	0.04 **	(0.52–0.98)
Age of Child (0 to 12 months-reference)	13–24 months	2.52	0.006 ***	(1.31–4.87)
	25–36 months	8.43	0.000 ***	(4.45–16.03)
	37–48 months	1.16	0.57	(0.69–1.91)
	49–60 months	0.94	0.82	(0.56–1.58)
Birth Order Number (Birth order 1-reference)	2 or 3	0.80	0.29	(0.54–1.21)
	4 or 5	0.47	0.000 ***	(0.31–0.71)
	6 or above	1.18	0.56	(0.67–2.07)
Mother’s BMI (less than 18.5 kg/m^2^-reference)	Equal to or more than18.5 kg/m^2^	1.45	0.08 **	(0.95–2.21)
Education of mother (illiterate-reference)	Primary	0.86	0.81	(0.24–3.04)
	Middle	0.85	0.84	(0.22–3.31)
	Matric and higher	0.32	0.14	(0.07–1.46)
Education of father (illiterate-reference)	Primary	0.67	0.54	(0.19–2.35)
	Middle	2.03	0.32	(0.51–8.11)
	Matric and higher	5.99	0.31	(0.61, 59.29)
Mother’s Working Status (Working-reference)	Not working	3.14	0.09 **	(0.81–12.18)
Working Status of Father (Govt. Job-reference)	Daily wage or labor and agriculture	0.52	0.46	(0.09–2.95)
	Private job	0.18	0.05 **	(0.04–0.96)
	Own business	0.05	0.002 ***	(0.007–0.35)
Household Deprivation Status (HDS-1-reference)	HDS-2	0.05	0.027 **	(0. 005–0.879)
	HDS-3	0.04	0.000 ***	(0.008–0.193)
Financial Stress (High-reference)	Medium	0.10	0.009 ***	(0.018–0.567)
	Low	0.006	0.000 ***	(0.005–0.061)
The model’s overall significance				
Number of Observations = 930		Prob > Chi^2^ = 0.0000		
LR Chi^2^ (26) = 231.56		Pseudo R^2^ = 0.1910		

Source: Authors’ estimation References: odds ratios; confidence intervals. R: reference category in the model. Significance level: *** if *p* < 1%; ** if *p* < 5%.

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
