# Peer review of "The Effects of Financial Stress and Household Socio-Economic Deprivation on the Malnutrition Statuses of Children under Five during the COVID-19 Lockdown in a Marginalized Region of South Punjab, Pakistan"

_children, 2022, doi:10.3390/children10010012_

Round 1

Reviewer 1 Report

Dear Sir/Mam

Please find bellow the requested review regarding the manuscript. The article contains a lot of useful information on the issue. The topic is very interesting and but use of sources is not appropriate. Although it has some useful information there are less references and the statements are not established. I suggest the authors to write more information with references.

The article contains a lot of useful information on the issue. It is quite clear what is already known about this topic and the research question is clearly outlined. The abstract is too brief and introduction section involves too many information. The research question is not justified clearly, given what is already known about the topic. The results are not discussed from multiple angles and conclusions answer the aims of the study partially. The conclusions are partially supported by references or results and the limitations of the study fatal and it is questionable if there are opportunities to inform future research. Positive: There are some strengths of the article that could have an impact in the field, such as the topic and its impact on the existed literature.

Author Response

Comments by Reviewer: The research question is not justified clearly, given what is already known about the topic.

Response by Author: Thanks for your valuable comments. First, this study observed the impact of financial stress created by COVID-19 during lockdowns and household socio-economic deprivation in a rural and deprived region of Pakistan. Previous research recently conducted in the context of COVID-19 has mainly observed that mental health-related issues due to lockdown or financial stress was associated with mental health and depressive symptoms among women during the COVID-19 lockdown. While in general there was almost limited/minor work on the effect of financial stress on malnutrition status during the COVID-19 lockdown. In Pakistan, whether COVID-19 played any role to increase malnutrition or not. In few studies conducted in other countries during COVID-19 concluded that COVID-19 has stopped the important public maternal and child health programs during lockdowns. So, it is hypothesized that the COVID-19 lowdown has increased the financial stress in families and further financial stress and socio-economic deprivation have increased the malnutrition status in under five children. Therefore, considering this vast research gap, this study investigates the effect of financial stress and household deprivation status on the malnutrition status of under-five children during COVID-19 in one of the most deprived regions of Punjab in Pakistan. (See lines 101-105)

Comments by Reviewer: The results are not discussed from multiple angles and conclusions answer the aims of the study partially. The conclusions are partially supported by references or results and the limitations of the study fatal and it is questionable if there are opportunities to inform future research.

Response by Author: Thanks for your valuable comments. In our understanding, your comment is about the summaries of results in the conclusion section and strengthening the limitation section. We have revised our discussion, conclusion, and limitation of the study which clearly explains our findings and objectives.

The addition is the first paragraph of the discussion: This study constructed the index of financial stress created by the COVID-19 lockdown in Pakistan. Furthermore, it observed the impact of financial stress and household socio-economic deprivation during the COVID-19 lockdown on the nutritional status of under-five children in one of the marginalized regions of South Punjab, Pakistan. The results of the study depicted that the prevalence of underweight is 41.89% and stunting is 58.86% in division Bahawalpur during the COVID-19 lockdown. Furthermore, descriptive results illustrate that 2.55% and 88.7 % of malnourished children were belonging to the HDS-1 and HDS-2 households category which shows the deprived segment of society. Moreover, 91.84% of malnourished children belong to those households that were facing high financial stress during the COVID-19 lockdown in the Bahawalpur region. (See lines 389-398)

Revised conclusion: The study assessed the effect of financial stress and household socioeconomic deprivation on the nutritional status of under-five children during the covid-19 lockdown. The results of the study depicted that the prevalence of underweight is 41.89% and stunting is 58.86% in division Bahawalpur during the COVID-19 lockdown. The logistic regression results depicted that gender, birth order and age of the child, BMI and working status of the mother, and father or parental working status are the other significant determinants of child malnutrition status. While the results of policy variables of our study (Financial stress and household deprivation status) showed that the probability of the child being malnourished decreases with the rise in the social and economic status of the household. Similarly, the probability of child malnutrition decreases when a household shifts from high financial stress to medium financial stress and then low financial stress. In sum, our findings concluded that household socioeconomic deprivation and financial stress created by the COVID-19 lockdown have impacted the nutritional status (17.26% stunting and 12.29% underweight increased). Socioeconomic deprivation in downgraded areas contributes to malnourishment, which might be eliminated with equal possibilities for human growth and increased funding for disadvantaged groups in less developed areas in form of social safety nets. Moreover, giving job security, especially during unexpected situations such as covid-19 will not only help people to feed their families but also overcome mental health issues.

The addition is in limitations and strengths of the study: The study assesses the impact of household socio-economic deprivation and financial stress on child malnutrition which was specifically shaped by COVID-19 during the lockdown in rural areas of southern Punjab, Pakistan. The data is collected from rural areas of three districts with 1152 households. So, for policy purposes, the results and scope of this study could be generalized for the whole of rural Punjab, Pakistan. The limitation of this study is that the questions developed to design the financial stress index were related to stress and job loss issues that rises during COVID-19. Moreover, this study used prearranged set of variables because of the importance of COVID-19 time to meet the objectives. The future directions could be how COVID-19 distrusted the essential maternal and child health and child nutrition programs in lockdown. Moreover, researchers could also observe the post-COVID-19 analysis for the malnutrition prevalence and which factors played the role in overcoming malnutrition in post-COVID-19. (See lines 521-532)

Reviewer 2 Report

In the introduction section, please provide data on the economic growth (GDP), unemployment rate, and inflation rate.

Please provide a literature review section.

There is no hypothesis in this study. 

What is the underpinning theory of this study?

It is not clear, how many districts, sub-districts, and union councils are in the division of Bahawalnagar? What are the criteria for selecting the respondents of the study?

There is inconsistency in the information about the data collection period- abstract (November 2020 and April 2021) and method section (July 2020 and November 2020). Please clarify.

The stress levels are classified into three levels: low (0-3), medium (4-7), and high (8-10). Any source of reference from the previous study?

The author should clearly discuss the policy implication of the study.

Author Response

Comment-1: In the introduction section, please provide data on the economic growth (GDP), unemployment rate, and inflation rate.

Response by Author: Thanks for your valuable comments. According to Pakistan Bauru of Statistics, the GDP growth rate was 6.03%, the unemployment rate was 4.35%, and inflation was 9.5% in 2021 (World Bank Data). (See lines 91-93)

Comment-2: Please provide a literature review section.

Response by Author: Thanks for your valuable comments. There is an extensive literature review in the Introduction Section which relates to the issue and justifies the problem regarding health and covid-19 as we are seeing child nutrition in the time of covid-19. Literature on covid and child health is adequate to present in our paper rather than adding an extra literature review section on socio-economic determinants of malnutrition. That would be extra writing and an increase in pages. To avoid extra writing we limit our literature in the introduction section concerning the topic justification.

Comment-3: There is no hypothesis in this study. 

Response by Author: Thanks for your valuable comments. It is hypothesized that the COVID-19 lockdown has increased the financial stress in families and further financial stress and socio-economic deprivation have increased the malnutrition status in under five children. (See lines 101-105)

Comment-4: What is the underpinning theory of this study?

Response by Author: Thanks for your valuable comments. This study is based on the household production function the framework of Becker (1965) and Strauss and Thomas (1995) [30-31], and the conceptual framework by Victora et al. (1997) [32-33].  (See lines 109-135)

Comment-5: It is not clear, how many districts, sub-districts, and union councils are in the division of Bahawalnagar? What are the criteria for selecting the respondents of the study?

Response by Author: The sample was distributed proportionally among 3 districts and 14 sub-districts (tehsils), and 42 union councils. In the division Bahawalpur, there are three districts (Rahymyar Khan, Bahawalpur, Bahawalnagar) and in these districts, there are 14 tehsils. We selected all rural households in three districts as the sample frame, and among these three districts, the rural households in all 14 sub-districts or tehsils as the sample frame in our study. The names of districts and tehsils are given in Table 1. Furthermore, a total of 129 UCs are included in all 4 tehsils of district Rahimyar Khan. There is a total of 88 UCs in all 5 tehsils of district Bahawalpur, and a total of 135 UCs are in all 5 tehsils of district Bahawalnagar. According to the Census (2017), around 75.36% population in division Bahawalpur is living in rural areas, and the number of rural UCs is more than the urban UCs. Therefore this study was conducted in rural areas of Division Bahawalpur. Due to cost and time constraints, we possessively selected only three rural union councils from each tehsil, and in this way total 42 union councils were considered for this study. The fixed 384 samples of each district were proportionately distributed from district to tehsils and from tehsils to UCs at the first stage. The households were selected randomly from UCs following the lady health worker’s register record in the second stage. (See from line 146-162)

Comment-6: There is inconsistency in the information about the data collection period- abstract (November 2020 and April 2021) and method section (July 2020 and November 2020). Please clarify.

Response by Author: Thanks for your valuable comments. This mistake is corrected in the method section the date is November 2020 and April 2021. (See lines 170-171)

Comment-7: The stress levels are classified into three levels: low (0-3), medium (4-7), and high (8-10). Any source of reference from the previous study?

Response by Author: Thanks for your valuable comments. This study followed a systematic process for developing a new index regarding financial stress specifically the COVID-19 situation. This index is developed in the same way as one study developed a mother’s health and nutritional literacy index (see reference number, 36), and another study developed the financial stress scale (see reference, 26; and reference number 72). This study gets help from these studies while developing the financial stress index.

Comment-8: The author should clearly discuss the policy implication of the study

Response by Author: Thanks for your valuable comments. We have discussed our policy implications regarding our two policy variables which are household socio-economic deprivation and financial stress. Moreover, we revised our conclusion according to your comment.

Conclusion: In sum, our findings concluded that household socioeconomic deprivation and financial stress created by the COVID-19 lockdown have impacted the nutritional status (17.26% stunting and 12.29% underweight increased). Socioeconomic deprivation in downgraded areas contributes to malnourishment, which might be eliminated with equal possibilities for human growth and increased funding for disadvantaged groups in less developed areas in form of social safety nets. Moreover, giving job security, especially during unexpected situations such as covid-19 will not only help people to feed their families but also overcome mental health issues.

Round 2

Reviewer 1 Report

I agree with the revised version.